# Effect of Low Zeolite Doses on Plants and Soil Physicochemical Properties

**DOI:** 10.3390/ma14102617

**Published:** 2021-05-17

**Authors:** Alicja Szatanik-Kloc, Justyna Szerement, Agnieszka Adamczuk, Grzegorz Józefaciuk

**Affiliations:** 1Institute of Agrophysics of Polish Academy of Sciences, Doswiadczalna 4 Str., 20-290 Lublin, Poland; a.kloc@ipan.lublin.pl (A.S.-K.); a.adamczuk@ipan.lublin.pl (A.A.); 2Faculty of Geology, Geophysics and Environmental Protection, AGH University of Science and Technology, 30-059 Krakow, Poland; jsze@agh.edu.pl

**Keywords:** clinoptilolite, soil, water, CEC, specific surface, zeolitization

## Abstract

Thousands of tons of zeolitic materials are used yearly as soil conditioners and components of slow-release fertilizers. A positive influence of application of zeolites on plant growth has been frequently observed. Because zeolites have extremely large cation exchange capacity, surface area, porosity and water holding capacity, a paradigm has aroused that increasing plant growth is caused by a long-lasting improvement of soil physicochemical properties by zeolites. In the first year of our field experiment performed on a poor soil with zeolite rates from 1 to 8 t/ha and N fertilization, an increase in spring wheat yield was observed. Any effect on soil cation exchange capacity (CEC), surface area (S), pH-dependent surface charge (Qv), mesoporosity, water holding capacity and plant available water (PAW) was noted. This positive effect of zeolite on plants could be due to extra nutrients supplied by the mineral (primarily potassium—1 ton of the studied zeolite contained around 15 kg of exchangeable potassium). In the second year of the experiment (NPK treatment on previously zeolitized soil), the zeolite presence did not impact plant yield. No long-term effect of the zeolite on plants was observed in the third year after soil zeolitization, when, as in the first year, only N fertilization was applied. That there were no significant changes in the above-mentioned physicochemical properties of the field soil after the addition of zeolite was most likely due to high dilution of the mineral in the soil (8 t/ha zeolite is only ~0.35% of the soil mass in the root zone). To determine how much zeolite is needed to improve soil physicochemical properties, much higher zeolite rates than those applied in the field were studied in the laboratory. The latter studies showed that CEC and S increased proportionally to the zeolite percentage in the soil. The Qv of the zeolite was lower than that of the soil, so a decrease in soil variable charge was observed due to zeolite addition. Surprisingly, a slight increase in PAW, even at the largest zeolite dose (from 9.5% for the control soil to 13% for a mixture of 40 g zeolite and 100 g soil), was observed. It resulted from small alterations of the soil macrostructure: although the input of small zeolite pores was seen in pore size distributions, the larger pores responsible for the storage of PAW were almost not affected by the zeolite addition.

## 1. Introduction

Zeolites are natural or artificial crystalline aluminosilicates exhibiting an open, highly porous structure containing cations balancing high electrostatic charge of the framework of silica and alumina tetrahedral units. The internal surface area of the zeolite framework can reach as many as several hundred square meters per gram [1]. These unique physicochemical properties of zeolites make them extremely effective cation exchangers, water sorbents and adsorbents of uncharged molecules, which, coupled with the abundance of zeolites in sedimentary deposits and in rocks derived from volcanic parent materials, have made them useful in many industrial, environmental and agricultural applications [2,3,4,5]. An analysis of the www-sites concerning world zeolite market [6,7,8,9] indicates that the annual production of zeolite is about 3 million tons. The main contributors are China, South Korea, Japan, Jordan, Turkey, Slovakia and the US. The global zeolite market size, of which natural zeolites account for around 30%, was valued at USD 2.9 billion in 2016 and USD 4.3 billion in 2019. It is expected to grow at a compound annual growth rate of 2.5–4.7% (depending on the forecasting institution). Agriculture is the main end-user of the total world production of natural zeolites (25%), and together with water treatment and air purification covers around 70% total demand. Within the agricultural sector, major applications concern animal husbandry (litter and fodder additives). Around 30% of the mineral is used as a soil conditioner with the belief that zeolite addition improves the physical and chemical properties of soil for a long time [2]. Zeolites are considered to improve many soil physicochemical properties. They increase soil infiltration rate, saturated hydraulic conductivity, water holding capacity, aeration and many others [3,4,5,10,11]. Various researchers reported that zeolite increases soil cation exchange capacity and water retention in the root zone [12,13], decreases mineral components leaching [5,14] and traps significant amounts of heavy metals and organic pollutants in contaminated soils [15,16]. Zeolite is not acidic but slightly alkaline and its use with fertilizers can help buffer soil pH levels, thus reducing the need for lime application [17,18]. Zeolites’ action as slow-release fertilizers are reported as well [3]. Unlike other soil amendments (e.g., lime), zeolite does not break down over time but remains in the soil to improve nutrient retention. Therefore, its addition to soil significantly reduces water and fertilizer costs by retaining beneficial nutrients in the root zone [4]. The above-mentioned effects of zeolites on soil properties are frequently used for explanations of their positive effects on plant yield, growth and survival of abiotic stresses (drought, metals toxicity, etc.) that have been reported in many papers for sunflower, soybean, tomatoes, radish, beans, potatoes, clover, maize, winter wheat, sugar cane and other plants [5,17,18,19,20,21,22,23,24,25,26]. These effects are achieved at different doses of the mineral: from less than 10 t/ha [27,28,29,30,31] up to 120 t/ha [32]. 

We believe that with low zeolite doses, at which the mineral dilution in soil is very high, changes in soil physicochemical properties should be insignificant and therefore their effect on plant growth and yield should be negligible. Assuming that the soil bulk density is 1.5 g/cm^3^ and the 15 cm soil layer weight is 2250 tons/ha, one ton of the zeolite dose per hectare is equivalent to an addition of 0.044 g of the zeolite to 100 g of the soil. 

To study this problem in more detail, we performed a field experiment to observe the reactions of a plant and a soil on low zeolite doses. Particular attention was placed on these physicochemical properties which are the most important for soil–plant interactions: CEC, specific surface area, water sorption energy, water retention and mesoporosity. According to the authors’ knowledge, similar literature reports are lacking. Because wheat is the most common cereal crop in Europe, a popular variety of spring wheat was selected as the testing plant, for which yield quantitative and qualitative parameters were measured. Wheat is very sensitive to water and nutrient deficits [33], and therefore a large impact of the zeolite addition was expected. From about 40 known types of natural zeolites, clinoptilolite is the most common, the cheapest and the most frequently applied [34]; thus, this mineral was used in the experiment.

## 2. Materials and Methods

A 3-year experiment (2014, 2015 and 2016) was conducted in Rogozno (51°38′ N, 22°56′ E, 168 m a.s.l.) on acidic brown soil of pH (KCl) = 4.9 and loamy texture containing 51% sand, 42% silt and 7% clay determined by laser diffractometry using Mastersizer 2000 Malvern UK apparatus according to the method described in detail by Ryzak and Bieganowski [35]. Soil organic matter level was 1.2% as determined according to Walkley and Black [36] and Nelson and Sommers [37]. The content of phosphorus was 9.8 mg P_2_O_5_/100 g soil that lies on a boundary between low and average phosphorus level. The low content of potassium was 7.7 mg K_2_O/100 g soil. This rather poor soil was selected with the belief that more pronounced effects of the further zeolite addition will be observed. 

Ground, 1 mm sieved zeolite material was prepared from a clinoptilolitic tuff deposited in Socirnica (Ukraine). The zeolite was purchased from Andalusia Ltd., Warsaw, Poland. The exchangeable cations content of the mineral were: 396 mmol/kg calcium, 377 mmol/kg potassium, 91 mmol/kg sodium and 58 mmol/kg magnesium, and its pH was 7.4. The exchangeable cations were measured by AAS in 1 M ammonium acetate extract, according to Chapman [38]. 

Scanning electron microphotographs of the studied zeolite using Phenom ProX desktop SEM (Thermo Fisher Scientific, Waltham, MA, USA) are presented in Figure 1.

The X-ray diffraction spectrum of the studied zeolite registered using Panalytical Philips X’pert PRO APD MPD XRD (PANalytical, Kassel, Germany) spectrometer, thanks to the courtesy of Professor Wojciech Franus from Lublin Polytechnical University, is shown in Figure 2, wherein characteristic clinoptilolite peaks are abbreviated by the letter C.

Nitrogen adsorption/desorption isotherms of the studied zeolite measured using 3Flex 3500 Surface Characterization Analyzer (Micromeritics Instrument Corporation, Norcross, GA, USA) are presented in Figure 3.

The content of clinoptilolite in the zeolite used is about 75%, and it was recognized from characteristic distances *d*_hkl_ = 8.95; 7.94; 3.96 and 3.90 Å. The other mineral components are opal, quartz and feldspars. Clinoptilolite occurs in forms of thin plates (around 10 µm), sometimes of distinct hexagonal shapes. The BET surface area derived from nitrogen adsorption was 26.1 m^2^ g^−1^ and the micropore volume (from t-plot) was 0.001 cm^3^ g^−1^. The average pore radius (from desorption isotherm) was 6.78 nm. The above data certify that the zeolite used was a high grade clinoptilolite. 

The zeolite was added to the soil in amounts of 1, 2, 4 and 8 t/ha. The rates of zeolite were chosen taking into account the financial possibilities of Polish farmers. The minimum price of one ton of a zeolite is around EUR 100. One ton of wheat costs around EUR 200. An average yield of wheat is around 4 tons per hectare. Even excluding the other costs of cropping, to apply 8 tons of a zeolite per hectare, a farmer must invest his whole income. 

Three replicates of a set of six 2 m × 10 m plots separated from each other by 2 m distance from each side were arranged on the experimental field. From each set, one plot was leaved as a control (abbreviated further as FZ0), four were fertilized with the above-mentioned doses of the zeolite (FZ1, FZ2, FZ4 and FZ8, respectively) and one was fertilized with nitrogen, potassium and phosphorus (FNPK). The plots within each replicate had random locations of different fertilization variants, as illustrated in Scheme 1.

The upper soil layer of the whole experimental field was mixed by 15 cm deep disking. The 3-year field experiment was designed to:-Compare NPK and zeolite action on soil and plants (first year of the experiment);-Look for eventual advantages of zeolite plus NPK over standard NPK fertilization (second year of the experiment). In this year we decided to apply NPK on all but FZ0 plots to prevent exhaustion of soil nutrients;-Look for the eventual presence of the long-term zeolitization effects (third year of the experiment).

In the first year of the experiment before sowing, all plots (including the control ones) were supplied with 150 kg/ha of slow-release nitrogen fertilizer Saletrosan^®^26macro (26% N and 13% S) to avoid plant growth differentiation by this macroelement. The FNPK plots were additionally enriched with 100 kg/ha of Agrafoska PK 20–30 (33% P_2_O_5_ and 30% K_2_O). This is worth noting that one ton of the zeolite contains 14.7 kg of exchangeable potassium (17.7 kg K_2_O); thus, 2 tons of the zeolite supplies a similar amount of potassium as 100 kg of Agorafoska. Although the above amounts are similar, the plant availability of exchangeable potassium (zeolite) and soluble potassium (Agorafoska) may be different. 

In the second year of the experiment, all but the control plots (only N fertilization, as in the previous year) were fertilized with NPK (100 kg/ha of Agrafoska PK 20-30 + 150 kg/ha of Saletrosan^®^ 26 macro). The NPK was applied to avoid exhaustion of the nutrients from the poor soil on which the experiment was performed and to observe eventual effects of combined zeolite/NPK treatment. It was also suspected that zeolite will accumulate some nutrients added as NPK and then act as the long-term fertilizer, as this is stated in the literature [3,4,5]. 

In the third year of the experiment, the soil fertilization mode was the same as in the first year. We expected that in this year of the experiment, the long-term effect of the zeolite already present in the soil will be exhibited. The general view of the fertilization treatments in particular years of the field experiment is summarized in Scheme 2.

In all 3 years, at the beginning of April, 190 kg/ha of spring wheat (*Triticum aestivum* L) cv. Izera were sown on the same plots in 10 cm rows by a seeder. At the wheat BBCH 1 (12) phase [39], 300 L/ha aqueous suspension containing 0.8 L of Puma Universal 069 EW were applied, and in the BBCH3 (31) phase, 400 L/ha 0.5% suspension of AMINOPIELIK SUPER 464 SL herbicides were applied. Whole plants were harvested in the first days of August from three randomly selected 0.5 m^2^ areas located on each experimental plot at least 30 cm from its boundary. The number of ears per square meter (NE); number of florets in the ears (NFE); average length of straw (LS) and ears (LE); weight of plant (WHP), straw (WHS) and ear (WHE); number of grains per ear (NG); weight of 1000 grains (WTG); grain moisture (MG); volumetric grain density (D); total protein content in grain (TPC) (according to Polish standard PN-EN ISO 20483: 2007); wet gluten percentage (GW) and gluten index (GI), PN-A-74042/02: 1993); falling number (F) (PN-EN ISO 3093: 2010); and grain yield (YG) were measured. 

Daily rainfall and average temperatures (average from the data at 8.00 and 18.00) were recorded during the growing seasons.

Soil properties were studied in two ways. In the first way (called later field soil experiment), soil samples taken directly from the experimental field, control plots (FZ0) and maximally zeolitized (FZ8) plots were examined. Roughly 1.5 kg of the soil samples were taken from 0–15 cm depth layer, from the centers of all 0.5 m^2^ areas from which the plants were harvested, 2 mm sieved and homogenized. Here, the soil parameters were measured in one replicate for each soil sample (for each treatment it was 1 sample from each of 3 areas from a given plot ×3 replicates of each plot). In the second way, (called later laboratory soil experiment) an average soil sample (soil taken from all control plots mixed together) with much higher doses of zeolite—0, 1, 4, 8, 20 and 40 g of the zeolite per 100 g of the soil—was examined. These soil–zeolite mixtures (abbreviated further as Z0, Z1, Z4, etc.) were carefully homogenized by hand and subjected to five drying/wetting cycles. Here, each parameter was measured in three replicates. 

The following parameters were determined in both field soil and laboratory soil–zeolite mixtures, as well as for the zeolite itself:pH in KCl (1:5 soil:solution ratio);Cation exchange capacity, CEC, at pH = 8.2 using Ba^+2^ as index cations (Mehlich method);Soil variable surface charge (called also pH-dependent charge), Qv (μmol g^−1^), measured from potentiometric titration curves registered under nitrogen atmosphere using an auto-titrator SM Titrino 702 (Metrohm, AG-Switzerland). The suspensions of the studied material in 1 mol·dm^−3^ NaCl solution were adjusted to pH = 2.95 (not changing within 5 min) and slowly titrated to pH = 10 with 0.1 mol·dm^−3^ NaOH. The amount (Mole) of the base consumed by the whole suspension, Nsusp, was used for neutralization of acidic groups of the solid surface, NS, and the acids present in the supernatant, Nsol. The NS value (NS = Nsusp-Nsol) measured between any two pH values is equivalent to variable surface charge developed by a soil in the given pH range. It is responsible for changes of soil CEC with changes in soil reaction and for a part of soil buffering capacity. More details on the method and calculations are given in Józefaciuk et al. [40];Amount of gravitational, GW, and plant available, PAW, water from selected points of water moisture versus water potential curves (called water retention or pF curves) were measured according to the procedure described in Richards [41] and Mullins et al. [42]. For water potential (pF) measurements, tensiometers placed in the soil were used and the soil moisture was measured with TDR hygrometer [43]. The pF is defined as a logarithm of a pressure necessary to remove water from soil pores (macropores). The pressure, understood as a water suction (F), is expressed as water height (in cm). The water retention curve provides the best characteristic of soil water storage at high moistures. Based on the above curve, one can distinguish different kinds of water stored in the studied medium. Gravitational water (GW) stored between pF = 0 and pF = 2.2 can easily flow down the soil profile under gravitation force and it is generally not used by plants. Water available for plants (PAW) occurs between soil water potential corresponding to pF = 2.2 (field water capacity, FWC) and pF = 4.2 (permanent wilting point);Mesopore volume, V (cm^3^ g^−1^), and average mesopore radius, R (μm), measured by mercury intrusion porosimetry using Micromeritics Autopore IV 9500 (Norcross, GA, USA) porosimeter. Pores detected by mercury intrusion porosimetry (MIP) belong to the range between 3 nm and 200 µm, roughly. The MIP measurements were carried out for 8 mm diameter and 8 mm height cylindrical aggregates prepared from the homogenized soil material and subjected to five wetting–drying cycles to stabilize the structure. The volume of mercury intruded into the aggregate at the maximum pressure was assumed to be equal to the mesopore volume. The average mesopore radius was calculated from pore size distribution functions obtained as the porosimeter reports. Details on porosimetric studies are described in Sridharan and Venkatappa Rao [44]. Mesopores play a crucial role in the formation of soil structure. They govern soil water, air and solute transport, soil compaction, aeration, root growth and many others;Surface area, S (m^2^ g^−1^), and water vapor adsorption energy, E, estimated from water vapor adsorption/desorption isotherms for the soil aggregates (as in MIP). The isotherms were measured by weighing the samples after stepwise equilibration at different relative water vapor pressures, p/p_0_, at 20 °C. The surface area was calculated from the linear form of the standard BET equation [45]. The average adsorption energy, E, was calculated from the energy distribution function, f(E), derived from adsorption isotherms plotted in energy coordinates, assuming that adsorption energy at a given p/p_0_ equals to ln(p_0_/p). More details on the calculations are given in Józefaciuk et al. [46]. The water vapor adsorption isotherm provides the best characteristic of soil water content at low moistures that are most often met in upper soil layers at normal weather conditions. The surface area expresses the summary surface of all soil particles. Particularly high input to the surface area are clay minerals, amorphous phases and organic matter. Except for water binding, surface area is responsible for sorption of humic acids, pesticides and herbicides, immobilization of contaminants and soil catalytic properties. It frequently correlates with soil CEC. The water adsorption energy reflects water binding forces. Systems with higher adsorption energy may grasp water from systems with lower adsorption energy.

## 3. Results and Discussion

### 3.1. Weather Conditions

The climatic data on the experimental site during the experiment are shown in Table 1 in comparison to long-term data. It is worth noting that the rainfall in April and particularly in May 2014 markedly exceeded the long-term data.

### 3.2. Effect of Zeolite on Plants

The yield structure of the studied wheat is characterized in Table 2.

In the first year of the field experiment, zeolite addition to the soil (along with nitrogen fertilizer) significantly increased the spring wheat grain yield per hectare in respect to the control plots. The yield at the highest zeolite dose (8 t/ha) was almost as high as that of NPK fertilization. The largest input to this increase had the number of ears per 1 m^2^ and the number of grains in ears. The latter parameter increased by 58% for the maximum dose of zeolite and by 78% for the NPK fertilization in relation to the control variant. Similar results regarding the positive effects of zeolite on maize, wheat, sugar cane and/or oats were recognized by Kisic et al. [22], Cairo et al. [28] and Szatanik-Kloc et al. [31]. Similarly to our studies, Fotyma and Fotyma [47] observed that the weight of 1000 grains did not affect the yield of winter wheat grain. However, Nouri-Ganbalani et al. [48] observed a close relationship between the yield and the weight of 1000 grains and the number of stems with ears. The cereal grain yield shows generally a high variability due to diverse influences of fore crop, thermal conditions, precipitation, soil conditions and agrotechnical measures, as well as genetic properties of the species, so various authors may present different opinions on the components influencing grain yield of cereals [49,50]. Generally, the best quantitative and qualitative parameters of wheat yield were observed in NPK-fertilized objects, similar to that of 8 t/ha zeolite fertilization, which is consistent with Aaina et al.’s [26] report for maize. In the second year of the field experiment, all plant parameters were the worst on FZ0 plots, which were fertilized only with nitrogen. On all other plots, the plant parameters were similar, regardless of the presence and the amount of the zeolite. In the third year of the experiment, the plant parameters were the best on NPK-fertilized plots and the worst on FZ0 plots. On all other plots, regardless the dose of the zeolite, the plant parameters appeared to be similar. 

Effects of fertilization on grain quality parameters are summarized in Table 3. 

In the first year of the experiment, grain volumetric density, falling number and the amount and quality of gluten significantly differed in various fertilization variants. The weight of 1000 grains was the highest in NPK variant and it practically did not differ in the other plots. However, Jelic et al. [29] reported that NPK fertilization did not affect the weight of 1000 grains. Slightly higher grain volumetric density was noted on the control than on the other plots. A decrease in volumetric density of the grains with increasing zeolite doses was noted. It indicated that the seed coat was weaning from the endosperm and worse filling of the caryopses occurred; thus, smaller grains of worse sowing value were produced. Zeolite had a positive effect on the falling number, which was the highest in traditionally NPK-fertilized plots and plots with the maximum dose of zeolite. There was no effect of zeolite fertilization on the total protein content, which is consistent with our previous findings for oats [28]. Similarly, Ozbahce et al. [32] observed no effect of zeolite on protein content in beans. Jelic et al. [29] reported that NPK fertilization did not affect the protein content in oat grains. In contrast, Sepaskhah and Barzegar [30] reported that the use of nitrogen and zeolite increased protein content in rice. The grain protein content and quality are determined by the supply of plants with nitrogen and phosphorus—the main component of nucleic acids, as well as potassium and magnesium, which activate enzymes involved in protein synthesis [51,52]. The qualitative and quantitative parameters of gluten improved after the application of zeolite and NPK. The lowest amount of wet gluten was observed in the control variant. No significant differences in the above parameter were observed in the other plots; however, a tendential increase in wet gluten with the zeolite rate can be seen. The gluten index reached a maximum at the highest dose of zeolite. The positive effect of zeolite on the amount and quality of gluten, a protein that determines flour quality parameters, observed in our experiment could result from better distribution of nitrogen and potassium in soil in the presence of zeolite, as postulated by Abdi et al. [27] and Ozbahce et al. [32]. The improvement of gluten properties increases the commercial value of the grains due to better bakery parameters of the flour. The decisive factor for the flour quality is also the falling number. The increase in this parameter in the fertilized objects may result from the limitation of the activity of amolytic enzymes, which is associated with smaller sprouting of grains. 

It was expected that that the presence of the zeolite will influence the plant yield and grain quality parameters also in the second and the third years of the experiment due to a long-term effect of soil zeolitization, which has been mentioned in many previously cited papers. Some differences were observed between the variants with zeolite and the control soil (FZ0) in terms of WHP, WHS, WHE, LE and FN in the third year of the experiment, which can be caused by an exhaustion of the control field from cationic nutrients or some important microelements that were taken by plants and removed with the harvest. However, the most important information is that the general effect of zeolite on plant growth and yield parameters was principally not observed in the second and the third years (regardless of the amount of zeolite present in the soil, the plant and grain parameters were similar). We think that the latter finding could be due to high amounts of rainfall in the first year of the experiment: if the soil was fully saturated with water, rapid ion exchange reactions [53] between the zeolite and the soil occurred, and the native cations present on the zeolite surface could be replaced mainly by multivalent soil aluminum and/or calcium cations, and the ionic composition of both exchange complexes became equalized. The temporary excess of nutrients replaced from the zeolite into the solution is either taken by plants or leached out the rhizosphere. Therefore, a favorable effect of any nutrients brought by the zeolite could be observed only in the first year of the experiment. In the next years, the excess of each cationic nutrient on the zeolite disappeared and it could act no more as a long-term fertilizer: zeolite supplied the same ions as the soil did. One might also suspect that the zeolite could accumulate some potassium added in the second year of the experiment and release it in the third year (to check it, the experiment with full NPK fertilization in the second year was designed); however, the above effect was also absent.

### 3.3. Effect of Zeolite on Soil 

#### 3.3.1. Field Soil

Physicochemical properties of the field soil sampled from the control and the maximum zeolite dose plots are shown in Table 4, along with the properties of the zeolite itself. 

Although the studied soil had very low cation exchange capacity, in our field experiment no effect of the added zeolite (up to 8 t/ha) on soil CEC was observed. However, an increase in CEC after zeolite addition has been frequently reported [7,47]. Abdi et al. [27] showed that the amount of exchange K+, Ca^2+^, Mg^2+^ in the soil (that can be roughly considered as the CEC) had increased significantly due to the use of 10 t/ha zeolite. Ravali et al. [54] observed around a 30% increase in CEC of a sandy loam after addition of 7.5 t/ha zeolite. Fudlel et al. [55] reported the positive effect of 5 t/ha zeolite on CEC of an Alfisol. Chomczynska et al. [56] observed around a 20% increase in CEC after 1%w/w zeolite addition to two degraded soils (edge of sand mine excavation and the fallow land).

The small effect of the zeolite on soil pH observed in the field studies is in line with Litaor et al.’s [24] results for peat soil after several months of a field experiment. Tallai et al. [25] also reported that in a 2-year experiment, the pH of zeolitized soil increased only by 0.1 unit. The above observations are, however, in contrast to the results presented by most researchers [5,16,17,57], who generally observed an increase in soil pH after zeolite addition. The small effect of the zeolite on soil pH may be connected with low buffering properties of the zeolite itself which can be concluded from the small variable charge of the zeolite, even smaller than that of the studied sandy soil. Since variable charge is responsible for pH buffering, soil amended with zeolite buffers less pH than the soil itself. In our field experiment, no effect of the added zeolite on the specific surface area and water holding capacity of the soil was observed. 

#### 3.3.2. Laboratory Soil Studies

Soil laboratory studies revealed a small influence on pH of the examined mixtures. At the maximum zeolite dose (Z40), the soil pH increased to 5.13. The cation exchange capacity of the studied zeolite was 127 cMol kg^−1^, around 20 times higher than that of the soil. The increase in CEC was proportional (R2 = 0.99) to the zeolite fraction, x, in the soil–zeolite mixture: CEC = 112.5x + 6.65 cMol kg^−1^. In the above equation, the proportionality coefficient reflects the CEC of the zeolite and the intercept reflects the CEC of the soil. Both above values coincide well with the measured values of both parameters measured for the zeolite and for the control soil. 

Dependencies of variable surface charge on pH calculated from potentiometric titration curves for the zeolite and the soil with different mineral doses are shown in Figure 4.

The total amount of the variable charge measured between pH 3 and pH 10 for the studied zeolite was 4.56 cMol kg^-1^, which is around four times smaller than the variable charge of the soil. High variable charge of the soil itself is most likely connected with high pH-dependent charge of soil organic matter created mostly by surface carboxylic groups of rather high acidic strength. Inorganic surface hydroxyls present on the zeolite surface are weaker and their dissociation at any pH is smaller, and thus less variable charge is created. The addition of zeolite decreases soil variable surface charge; however, contrary to the surface area, the dependence of CEC on the fraction of zeolite is not linear, i.e., the variable charge is not additive.

Water vapor adsorption isotherms on zeolite and the soil with different zeolite doses are shown in Figure 5.

In the whole relative pressure range, the adsorption is the highest for zeolite and the lowest for the control soil. The addition of zeolite increases water vapor adsorption on the soil. The studied zeolite has very large surface area, 125.5 m^2^ g^−1^. The addition of zeolite to the soil results in an increase in surface area proportional (R2 = 0.99) to the zeolite fraction, x, in the soil–zeolite mixture: S = 128.9x + 52.1 m^2^ g^−1^. The slope of the above equation reflects the surface area of the zeolite and the intercept reflects the surface area of the soil. Both above values coincide well with the measured values of both parameters measured for the zeolite and for the control soil. Low percentage of the zeolite added to the soil, even at its highest rate (8 t/ha), was a reason that no effects of soil zeolitization were noted in the field. It means that only very high amendments of zeolite can significantly increase soil specific surface. An increase in the specific surface area resulting from the addition of zeolite can be beneficial for an increase in organic compounds’ sorption capacity of the soil, as postulated by Cairo et al. [28]. Binding natural soil humic compounds limits their mineralization and loss of soil organic carbon [58]. From this point of view, even a small addition of zeolite to the soil can have a positive effect on the environment quality. Large surface area is also a crucial feature for high water sorption of the mineral; however, the adsorbed water is not used by plants. High water sorption by the zeolite may be dangerous for plants: dry zeolite can confiscate water from a rainfall coming after dry periods.

Water vapor adsorption energy of the zeolite, 4.98 RT (R is universal gas constant and T = 293 K is the temperature of the measurements) is practically equal to that of the soil, meaning that the zeolite and the soil bind water with similar forces. No difference in water binding between the zeolite and the soil indicates that at low moistures, where adsorption energy governs water binding, the zeolite of large specific surface can grasp huge amounts of water from the soil. 

Water retention curves for the studied soil–zeolite mixtures are shown in Figure 6. 

While zeolite can hold water up to 50% of its volume, the amount of plant available water is much less (20.9%) compared to soils of heavier textures. The amount of gravitational water (8.7%) is less than in the studied soil. The highest dose of zeolite decreases the GW of the soil from 11.27 ± 0.33% (Z0) to 9.33 ± 0,56 (Z40) and increases the amount of PAW from 9.48 ± 0.29 (Z0) to 13.1 ± 0.61% (Z40). As it is seen, even in laboratory experiments, high zeolite doses had rather small effects on gravitational and plant available water. Practically all of the literature reports are in contrast to these results. The increase in water in the soil in the available humidity zone due to the high porosity of the crystalline structure of zeolites was reported by Yasuda et al. [59] and Bernardi et al. [60]. According to Kralova et al. [7], soils enriched with natural zeolite were able to increase the water retention capacity by 18–19%, and for sandy soils even up to 50% [61]. More examples of the positive effect of zeolites on soil water retention are presented by Nakhli et al. [62] in their excellent review of the subject. Both gravitational and plant available soil water depend on soil macrostructure (large pores range, gravitational water) and soil mesostructure (medium pore range, plant available water) [63]. The slight effect of zeolite on plant available water is most likely connected with its small impact on soil macro- and mesostructure, which can be seen from the porosimetric measurements presented below. It is worth mentioning that around 20% of water bound by the zeolite (the part of moisture located above pF = 4.2 line in Figure 5) cannot be used by plants. This water occupies the finest pores present in the zeolite mineral network.

Pore size distribution functions derived from mercury intrusion porosimetry are shown in Figure 7. Not to shadow the lines, the results are presented only for the control soil, the zeolite and the soil–zeolite mixture with the maximum zeolite dose.

In the studied soil, practically only large pores are present, located around 3.2 µm (maximum frequency at log(R) = 0.5). The zeolite poses dominant pores around 6 µm (maximum frequency at log(R) = 0.8), a significant amount of medium-sized pores, around 0.032 µm (at log(R) = −1.5), and some very fine pores, around 0.002 µm (at log(R) = −2.7). Even the highest amendment of the zeolite affects the soil pore size distribution to a small extent: in the Z40 sample, the large pores peak is conserved and the frequency of finer pores occurrence increases slightly. The above results indicate that both the macro- and mesostructure of the soil-zeolite system were not significantly altered by the added mineral. We believe that this is due to the fact that the added zeolite had rather large grains. Its addition is equivalent to the enrichment of the soil with sand-size material that does not affect larger pores. Possibly, the large-grained zeolite addition to clayey soils may improve soil aeration due to soil texture coarsening. In our opinion, these results suggest that for an improvement in the retention of plant available water, zeolites of very fine granulation should be applied; however, high costs of grinding and difficulties in management may limit their application.

##### General Remarks

The significant effect of zeolite application on plant yields in the first year of the experiment does not seem to be caused by the improvement of soil physicochemical characteristics. As mentioned before, 1 t/ha zeolite dose is equivalent to the presence of about 0.04% of the zeolite in the soil (0.044 g zeolite per 100 g of the soil). Therefore, it is unlikely that this amount of zeolite can affect the physicochemical parameters of a soil because of very high dilution of the added zeolite in soil material. Very large rates of this mineral are required. For example, Ozbahce et al. [32] reported that soil CEC increased only after the addition of zeolite in a dose of 90 t/ha. The observed improvement of spring wheat yield in the first year of the experiment may result from the release of potassium, calcium, magnesium and some microelements to the root zone by the zeolite exchange complex. Because all field plots were fertilized with the same amounts of nitrogen, it should not differentiate the yield; however, the combined effects of zeolite and nitrogen might have been important in this respect, as stated by Hung and Petrovic [64], who observed that the use of zeolite improved the effectiveness of nitrogen application in the soil by about 16–22% due to better nitrogen distribution in the soil. Cairo et al. [28] also reported that zeolite plus nitrogen increased the yield of sugar cane by almost 80%. Sepaskhah and Barzegar [30] attributed the higher yield of rice grain to the improved N retention in the soil by the added zeolite. Zeolites can also increase the availability of phosphorus [26] and limit the mobility of toxic heavy metals [15].

It seems that the paradigm of the improvement of water storage after soil zeolitization is wishful thinking rather than a real fact—this was proven based on water retention curves for the laboratory studied soil amended with different zeolite doses and supported by porosity measurements.

## Data Availability

All data are presented in the paper.

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
