# Peer review of "Effect of Low Zeolite Doses on Plants and Soil Physicochemical Properties"

_materials, 2021, doi:10.3390/ma14102617_

Round 1

Reviewer 1 Report

The paper presents the effects of zeolite addition on soil properties and plant growth. This study is of interest since it contribute to a subject that, even was intensively studied is still under debate. From this point of view, the article is sufficiently novel and interesting to warrant publication. The manuscript contains all the key elements: title, abstract, introduction, materials and methods, results and discussions, conclusions, references, as requested in the instructions for authors. Nevertheless, there are necessary improvements to warrant the publication of manuscript:

  1. Title: Even is a subject under debate, in my opinion this title “ … truth and fiction revealed from field and laboratory experiments” is somewhat controversial, taking into account that many scientific publications were published on this topic. In my opinion it should be changed to eliminate “truth and fiction”.
  2. Abstract, page 1, lines 15-16: “an increase of spring wheat yield was observed, however any effect on soil properties was noted”. Please indicate on which soil properties since in the next sentence it is stated that it was an extra nutrients supply. The content of these minerals are not a soil chemical property of soil? The same for the line 23, “A lack of the zeolite effect on soil properties …” this is somehow in contradiction with other affirmations like “cation exchange capacity and surface area increased proportionally to the zeolite percentage in the soil. The pH-dependent charge of the zeolite was lower in that of the soil”. Indeed, maybe at the application of zeolite in low rate the changes are not significant, but it exists (proportionally to the zeolite percentage in the soil, as the authors stated). An affirmation like those in the last sentence of the abstract: “A paradigm that zeolite (in commonly applied rates) improves soil properties should be denied” is also controversial, and should be removed. Maybe, it can be affirmed that the changes are not significant, but not such a categorical affirmation since the scientific results validated by publication of other scientific works revealed these changes. Also throughout the text should be stated that no significant changes in soil (physical?) properties were observed in place of “lack the zeolite effect on soil properties”
  3. Material and Methods, page 2, lines 93, 94: the mineral contained 396 mmol/kg calcium; 377 mmol/kg potassium; 91 mmol/kg sodium; 58 mmol/kg magnesium. These concentrations are those extracted in 1M ammonium acetate or represent the total content? How these concentrations were measured? For zeolite characterization, other parameters such as composition in major oxides, loss on ignition, XRD pattern, SEM images, BET analysis are necessary to shows that the used material was clinoptilolite zeolite.

Reviewer 2 Report

A manuscript contains two data blocks. The first one presents the results of an experiment on the effect of low doses of zeolite on some soil properties, structure and quality of the wheat yield in a three-year field experiment. The second data block is devoted to the evaluation of the effect of high doses of zeolite on some soil properties under laboratory conditions. Main findings of the study are:

- at low application dosages (up to 8 t/ha), the positive effect of zeolite in the field is observed only in the year of application;

- under lab conditions at high application dosages (up to 40 g / 100 g) zeolite introduction resulted in increase in cation exchange capacity and surface area, but demonstrated low effect on plant available water and soil macrostructure.

Based on the obtained results the authors concluded “A paradigm that zeolite (in commonly applied rates) improves soil properties should be denied”.

When reading the manuscript, a number of crucial questions arise. They are:

  1. Field experiments were carried out with extremely low doses of zeolite. It is unclear what were the reasons to state that the introduction of zeolites in high doses does not have a long-term effect either.
  2. The scheme of the field experiment is very complicated and unclear. In the first year of the experiment, the control FZ0 differs from the zeolite variants only by one parameter, namely, the introduction of zeolite in increasing doses. Then nothing is clear with the second and third years of the experiment. In the second year of the experiment, the FZ0 variant (control of the first year of the experiment) differs not only in terms of the application of zeolite in the previous year, but also in terms of the application of phosphorus and potassium fertilizers in the current year. The authors, apparently, suggest using the FNPK variant as a control in the second year of the experiment, but this fact is not indicated. However, phosphorus and potassium were added to the FNPK soil in the first year of the experiment, but not to the soil with zeolite. So, it can be assumed that the soil of the FNPK variant was enriched with nutrients before sowing wheat in the second year of the experiment, while soils with zeolite were not. The latter means FNKP soil can not be used as a control. This assumption could be refuted by analyzing the content of mobile phosphorus and potassium in soils, but these results are not available in the manuscript.
  3. Conclusions about the absence of the effect of zeolites on soil properties are extrapolated on the basis of an experiment with 1 type of zeolite on 1 variety of soil. At the same time, the characteristics of the zeolite are not given in sufficient detail and it is not clear whether the studied zeolite is a typical zeolite or not. Although the authors talk about the typicality of zeolite (lines 82-83), there is no experimental evidence for this. On the other hand, the soil that the authors describe as "poor" is also insufficiently characterized. The name of the soil implies that it is "rich" rather than “poor” soil. Prefix Eutric (from Greek eu, good, and trophae, food) can be applied if an effective base saturation [exchangeable(Ca + Mg + K + Na) /exchangeable(Ca + Mg + K + Na + Al) of ≥ 50% (http://www.fao.org/3/i3794en/I3794en.pdf). It is these soils that are characterized by the highest productivity in agriculture. Besides, the soil texture is specified incorrectly. The authors write “silty-sand” texture, but the soil possess loam texture (https://www.nrcs.usda.gov/wps/portal/nrcs/detail/soils/survey/?cid=nrcs142p2_054167). However, it is well known that the maximum efficiency of zeolites is observed on sandy soils.
  4. Illogical conclusions and hypotheses. The main hypothesis of the manuscript is: “A hypothesis of the current studies was that if changes in soil properties really govern the plant reaction on the zeolite addition, no effect on plants would be observed at low zeolite doses at which the mineral cannot impact the soil properties”. According to the results, the authors did not observe the effect of zeolite on soil properties in all the years of the experiment. It is rather arguable, since the authors themselves claim that in the first year the introduction of zeolites led, apparently, to an increase in the content of potassium (and microelements) available to plants. This means that the chemical properties of the soil have changed. The beneficial effect of zeolites on wheat was noted in the year when the zeolite affected the soil properties (the content of available potassium and microelements). The effect of zeolites on soil properties and wheat growth was not observed in the second and third years, when zeolite did not alter studied soil properties. It would seem that these findings completely coincide with the hypothesis of the authors and it should be concluded that the positive effect of zeolite on the plants is due to the positive effect of zeolite on the soil. Nevertheless, the authors draw the opposite conclusion.

Particular remarks:

Lines 10-11. Zeolites are mainly used as a component of slow release fertilizers, rather than slow release fertilizer. Please correct.

Lines 16-17. There are no data on available K presented, so please re-write the sentence.

Lines 20-22. There is no evidence to support this statement in the manuscript.

Lines 32-33. The conclusion is not based on the data presented in the manuscript.

Line 40. Please add a Ref.

Line 55. Please add a Ref.

Lines 72-74. The hypothesis proposed by the authors is almost impossible to prove or refute, since it is formulated too generally. What specific soil properties should be evaluated? After all, it is impossible to study all the soil properties. For example, in this work, biological properties were left out of the scope of the study. There is no evidence that the properties chosen by the authors for the study are really decisive in this case.

Line 86. Please correct name of the soil texture.

Line 90. Eutric Cambisol are supposed to be highly productive soils. So, please check the name of the soil.

Lines 92-94. Please specify SiO2/Al2O3 ratio to characterize the sample of zeolite used. This parameter determines the number of cation-exchange sites.

Lines 110-125. The text is very difficult to understand. Please make a table where amendments for the variants would be specified by year. Please specify what was used each year as a control.

Lines 126-127. Please write a scientific name in italic.

Table 2. Please indicate a statistical approach used to compare averages. According to the presented data, differences were seemingly observed between the variants with zeolite and the control (FZ0) in terms of WHP, WHS, WHE, and LE in the third year of the experiment. However, this data is not discussed in the text.

  1. Please check the dosage of zeolite (10 t/ha).

Table 3. Please indicate a statistical approach used to compare averages. Please discuss influence of zeolite on the falling number in the third year of the experiment.

Lines 286-287. The authors' point of view is not clear. In this paragraph, they simultaneously postulate that potassium was washed out from the zeolite due to heavy precipitation, but it was the absorption of potassium from the zeolite by wheat plants that caused their better growth. Please re-write.

Line 279. Please add a Ref.

Line 281. Please add a Ref.

Table 4. Please indicate a statistical approach used to compare averages. Please use cmol instead of mmol for CEC.

Lines 303-318. Without specifying the doses at which the effects of zeolite on the soil were or were not observed, this discussion is very arguable.

Line 356. Was it 10 t/ha or 8 t/ha?

Figure 3. Water retention curve traditionally is a relationship between the water content and the soil water potential, not vice verse. Please use water content for x-axis and water potential for y-axis.

Lines 379-391. The authors write about the atypical effect of the used zeolite on the physical properties of the soil, which again raises the question of the typicality of the used zeolite.

Lines 391-399. This part of the manuscript is mainly speculative arguments that are not supported by either presented data or literature. Please exclude.

Figure 4. According to Ramesh and Reddy, 2011 (the authors cite this paper), zeolites possess pore diameter from 0.3 to 1.0 nm. According to the Figure 4, zeolite used in this work can be characterized with pore size of 5 mkm. Please explain.

Lines 410-415. The Figure description does not match the data presented in the Figure. In the Figure 4 the highest pore content with a logarithm(pore radius) of about 0.6-0.8 can be observed. However, the authors write about pores with logarithm(pore radius) of -1.5 and -2.4. Is this correct?

Lines 436-439. It is not clear from what data the authors draw this conclusion. The authors themselves repeatedly emphasize that they used low doses of application, which should not have a pronounced effect on the physical and chemical properties of the soil. However, the positive and prolonged effect of zeolites is observed at high doses of application.

Lines 439-444. Does not relate to the presented data. Please exclude. 

Round 2

Reviewer 1 Report

The authors responded adequately to my comments and I consider that the manuscript can be published. 

Author Response

Dear Referee 1,

Thank you for your opinion. Your valuable comments helped us to markedly improve the paper.

Yours sincerely, the authors.

Reviewer 2 Report

To improve the manuscript, the authors made significant efforts. However, there are still some crucial flaws. In addition, when answering comments, the authors did not specify lines in the manuscript. Therefore, even though the corrected fragments are highlighted in red, in several cases I could not find the corrected versions of the text. Please check a pdf-file with comments.

Author Response

Der Reviewer 2. Please see the attached document containing your comments along with our replies.

Yours sincerely, the authors.
